## Comment

 

**Author for correspondence:**
Allowen Evin
e-mail: allowen.evin@umontpellier.fr

# Building three-dimensional models before destructive sampling of bioarchaeological remains: a comment to Pálsdóttir *et al.* (2019)

Allowen Evin[1], Renaud Lebrun[1], Marine Durocher[1,2,3], Carly Ameen[4,5], Greger Larson[6] and Naomi Sykes[4]

[1]Institut des Sciences de l'Evolution-Montpellier, UMR 5554-ISEM, CNRS, Université de Montpellier, IRD, EPHE, 2 place Eugène Bataillon, CC065, 34095 Montpellier Cedex 5, France
[2]Laboratoire Archéozoologie et Archéobotanique: Sociétés, Pratiques et Environnements, (AASPE), Muséum national d'Histoire naturelle, CNRS, Alliance, Sorbonne Université, 55 rue Buffon CP 56, 75005 Paris, France
[3]Institut de Systématique, Evolution, Biodiversité (ISYEB), Muséum national d'Histoire naturelle, CNRS, Sorbonne Université, EPHE, CP51, 57 rue Cuvier, 75005 Paris, France
[4]Department of Archaeology, University of Exeter, Exeter, UK
[5]Department of Archaeology, Classics and Egyptology, University of Liverpool, Liverpool, UK
[6]The Palaeogenomics and BioArchaeology Research Network, Research Laboratory for Archaeology and History of Art, University of Oxford, Oxford, UK

AE, 0000-0003-4515-1649; RL, 0000-0002-5819-2653;
MD, 0000-0002-8279-4099; CA, 0000-0002-4580-2125;
GL, 0000-0002-4092-0392; NS, 0000-0001-6114-7557

## 1. Introduction

Pálsdóttir *et al.* [1] have recently brought attention of the bioarchaeology and archaeozoology communities, museums and other institutions that curate archaeofaunal collections to the importance of achieving reasoned destructive sampling strategies when performing analyses requesting destruction of bioarchaeological remains for e.g. C14 dating, ancient DNA, palaeoproteomics, collagen fingerprinting or isotope analyses. Though the results obtained from such approaches produce major advances in our understanding of past fauna and their relationship with humans, such approaches cannot (yet) be performed without the destruction of at least a small portion of bone or tooth. Though Pálsdóttir *et al.* [1] focus on animal remains, the same is true for plants with cereal grains, chaff, seeds, charcoal and wood that can be submitted to the same variety of bioarchaeological analyses. Pálsdóttir *et al.* [1] suggested that advances in three-dimensional (3D) imaging to build digital models can be used to record specimens before destructive sampling. Here, we provide information

about available techniques to build such archives. We also emphasize the importance of the petrous bone, currently the most targeted bone for ancient DNA studies [2,3].

# 2. Building digital three-dimensional models

The acquisition, sharing and archiving of 3D models is now widespread and democratized thanks to an improvement in technical solutions, an increase in the amount of equipment in laboratories, the growth of interdisciplinary research and drastic cost reductions. It is now possible to easily acquire low-cost 3D models of bioarchaeological remains before their destruction or degradation. Possibilities are of two main kinds: X-ray computed tomography (CT, such as the ones commonly used in hospitals) and micro-tomography (µCT, increasingly found in research laboratories) or surface scans (e.g. photogrammetry, or laser and structured light scanners).

## 2.1. Photogrammetry and surface scan

Photogrammetry is the cheapest and most accessible way to create a 3D model since it requires only a camera and software to build the models. Some software is free and/or accessible online (e.g. https://micmac.ensg. eu/ or http://ccwu.me/vsfm/), though the most widely used is Metashape (https://www.agisoft.com/) costing approximately 500 euros for an educational licence. Similar to photogrammetry, laser and structured light surface scanners reconstruct the external topology of the object. In terms of time, both approaches require working on a specimen for approximately 15 min, not including time to set up equipment (plus computer time for photogrammetry), but those methods only capture the external, visible, part of the object.

## 2.2. X-ray CT and µCT

X-ray CT produces 3D models of both internal and external bone structures. Though it requires access to specific (expensive) facilities and bringing specimens to the machine, this method provides high-resolution models at a very reasonable cost per specimen. For example, we digitized 271 rabbit calcanea in a single approximately 30 min scan using an easytom 150 µCT scan (www.rxsolutions.fr), at a resolution of 70 µm, which is sufficient for further analyses (e.g. geometric morphometrics) (figure 1). Many public X-ray µCT platforms charge less than 100 euros per hour (less than 50 euros in our case) which translates to less than 40 cents per bone (19 cents in our case). Depending on the resolution and the size of the specimens, this cost may change but would always be a fraction of the cost per sample for e.g. ancient DNA studies. The full 3D model reconstruction and data post-treatment will also require time to identify, label and save the separate models (less than a day in our case).

A recent paper demonstrates a correlation between accumulating X-ray dose higher than 2000 Gray and decreasing aDNA quantities, but no effect was detected for doses below 200 Gray [4]. In our rabbit bone case, we worked at approximately 1 Gray $h^{-1}$ (i.e. approx. 0.5 Gray for the 30 min duration of the scan) obtained by placing the specimens approximately 15 cm from the X-ray source of 70 kV and 33 µA, and using a 1.1 mm thick aluminium filter (ideal for archaeological material though other filters exist, value obtained using dosimex 2.0 [5]).

Thus, because X-ray dose is cumulative, caution should be used if multiple scans have to be performed on the same specimen (e.g. for multiple studies) and a possible safe recommendation would be to avoid scanning a given archaeological sample more than five times and to record the number of the experiment conducted as well as the setting parameters used to perform the scan.

Storing data has a cost (50 Go of raw and derived data per hour in our case) which may be a limiting constraint if a large collection of 3D models has to be maintained.

# 3. Research potential of petrous bones and the inner ear

The petrous bone is now the first choice for many ancient DNA analyses because its density means it more often contains well-preserved DNA [2,3]; however, because of the small size of the petrous bone, it is frequently completely destroyed during destructive analyses [1].

The petrous bone is part of the temporal bone which is itself part of the endocranium and contains the component of the inner ear. Pálsdóttir *et al.* [1] pointed out that they can be used for fetal ageing, sexing and dietary isotope studies in archaeozoological analyses, but it has also been extensively studied in palaeontology and biology (e.g. [6,7]). The inner ear is responsible for sound detection and balance in vertebrates and therefore carries unique information about living animals. For example, a recent study

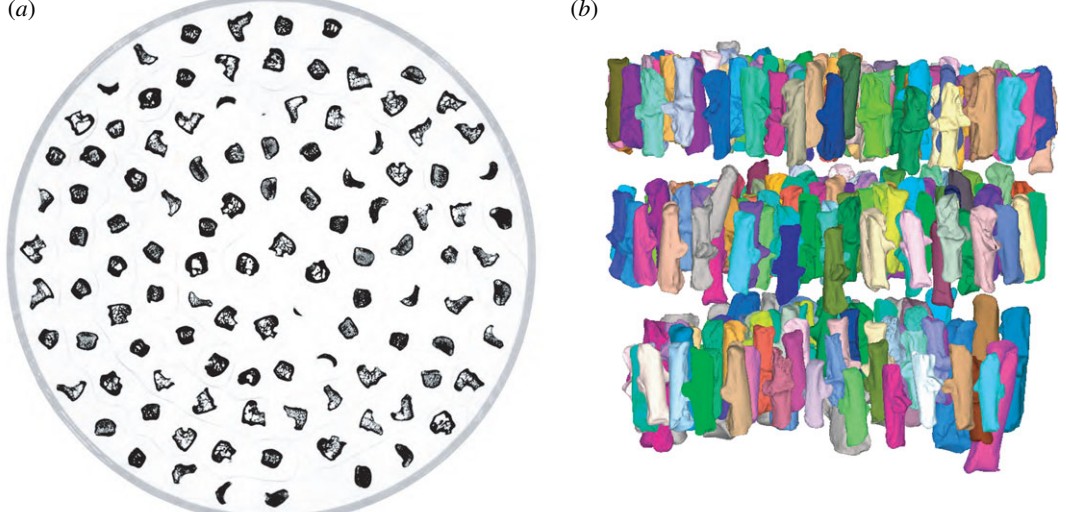

**Figure 1.** Visualization of 3D data acquisition of 271 rabbit calcanea. The bones were positioned in a 10 cm diameter tube in three layers before being µCT scanned using an easytom 150 (RX-solution) scanner. (*a*) µCT virtual slice through one of the three bone layers. (*b*) Visualization of the 271 3D surfaces, one colour per bone (https://morphomuseum.com/morphodig).

on wolves, dingoes and domesticated dogs [8] explored the use of the inner ear shape as a marker of domestication, a subject of direct interest for the archaeozoological community.

## 4. Conclusion

Archaeological remains are cultural and biological heritage that have to be managed with great care. No one can predict the future of bioarchaeological methods, and the value and uniqueness of remains from animals and plants should be assessed prior to destructive sampling. It is likely that in the future, the samples needed for destructive analysis will be smaller and smaller, so that one day one can render them completely non-destructive. In the meantime, archiving models and pictures of damaged archaeological remains is an easy and cheap alternative. If all petrous bones used for aDNA studies over the last years had been CT scanned before destruction, we would have had access to an amazing collection with bioarchaeological significance. It would have been possible, for example, to perform a comparative study of hearing evolution during domestication and the possibility of using inner ear morphology as a domestication marker which alongside the compatible DNA analyses would have been invaluable for the advancement of archaeozoological research.

Data accessibility. This article has no additional data.

Authors' contributions. A.E., R.L. and M.D. carried out the CT scan of the rabbit bones and drafted the manuscript. R.L. carried out the CT scan measure analyses. C.A., G.L. and N.S. prepared the specimens and conceived the study. All authors gave final approval for publication.

Competing interests. The authors have no competing interests.

Funding. Not relevant.

Acknowledgements. We acknowledge the MRI platform member of the national infrastructure France-BioImaging supported by the French National Research Agency (ANR-10-INBS-04, 'Investments for the future'), the labex CEMEB (ANR-10-LABX-0004) and NUMEV (ANR-10-LABX-0020).

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
