## [Reviewer comments · Royal Society Open Science]

Review History

RSOS-192034.R0 (Original submission)

Review form: Reviewer 1 (Albina Palsdottir)

Is the manuscript scientifically sound in its present form?

Yes

Are the interpretations and conclusions justified by the results?

Yes

Is the language acceptable?

Yes

Do you have any ethical concerns with this paper?

No

Have you any concerns about statistical analyses in this paper?

No

Recommendation?

Accept with minor revision (please list in comments)

Comments to the Author(s)

Paper review:

The paper "Building three-dimensional models before destructive sampling of bioarchaeological remains: a comment to Pálsdóttir et al (2019)" provides clear and valuable information about the costs and time needed to make 3D models of bioarchaeological remains before destructive sampling and will be a useful resource for scientists in the field. The paper should be accepted with minor revisions.

Page 1

Line 27: The paper "Not a limitless resource: ethics and guidelines for destructive sampling of archaeofaunal remains" addresses more than just the archaeozoological community but also museums and other institutions that curate archaeofaunal collections as well as the disciplines doing destructive sampling such as the ancient DNA community. You might want to change the first line of the paper to reflect that.

Page 2

Line 3-4: The text is a bit unclear, I suggest changing it to something like this:

In terms of time, both approaches require working on a specimen for ~15 minutes, not including time to set up equipment (plus computer time for photogrammetry) but those methods only capture the external, visible, part of the object.

Line 18-19: The text is a bit unclear, I suggest changing it to something like this:

Depending on the resolution and the size of the specimens, this cost may change but would always be a fraction of the cost per sample for e.g. ancient DNA studies.

Line 22-24: The text is a bit unclear, I suggest changing it to something like this:

A recent paper demonstrates a correlation between X-ray dose-levels higher than 2000 Gy and decreasing aDNA quantities but no effect was detected for doses below 200 Gy [4].

Line 24: You talk about Gy and Gray, I am not familiar with this measurement unit, is it the same? If so the same spelling should be used if not the difference needs to be explained.

Line 33: I suggest changing this to

Petrous bones are now the first choice for many ancient DNA analyses because its density means it more often contains well preserved DNA [2,3], however because of the small size of the petrous bone, it is frequently completely destroyed during destructive analyses [1].

Review form: Reviewer 2 (David Orton)

Is the manuscript scientifically sound in its present form?

Yes

Are the interpretations and conclusions justified by the results?

Yes

Is the language acceptable?

Yes

Do you have any ethical concerns with this paper?

No

Have you any concerns about statistical analyses in this paper?

No

Recommendation?

Accept with minor revision (please list in comments)

Comments to the Author(s)

This is an unusual response paper, in that it's not reacting against anything in the original article, but rather expanding usefully on a point that Palsdottir et al make in passing - namely the potential of 3D scanning of zooarchaeological bones prior to destructive analysis. As such, it doesn't represent an essential correction or counterpoint, but is nonetheless a highly valuable contribution that is best read alongside the original paper.

I have only two suggestions for changes, both of which are minor:

1. Talking about costs is useful, if likely to become out-of-date fast. However, it seems a little odd to switch between \$ and €. I would recommend standardising currencies to one or the other.

2. In the conclusion you refer to destructive sample requirements becoming smaller and smaller until eventually they are not destructive at all. While it's theoretically possible that new technologies might permit entirely non destructive analyses in future, this would surely require lateral developments over and above the expected reduction in sample size requirements. I'm not sure such advances can be considered 'likely'. Hence I'd suggest replacing 'completely' with 'virtually'.

(nb. some extant "non-destructive" methods, e.g. the ambic soaking protocol for ZooMS, are in reality destructive, despite not having an immediately visible impact on sample morphology. Future 'non-destructive' methods should likewise be treated with scepticism until their long-term curatorial impacts are assessed)

Review form: Reviewer 3 (Trish Biers)

Is the manuscript scientifically sound in its present form?

Yes

Are the interpretations and conclusions justified by the results?

Yes

Is the language acceptable?

Yes

Do you have any ethical concerns with this paper?

No

Have you any concerns about statistical analyses in this paper?

No

Recommendation?

Accept as is

Comments to the Author(s)

Thank you for a straight-forward assessment of 3D modelling and scanning techniques that can be used for zooarchaeological remains. The emphasis on loss of data from remains not imaged prior to destruction is crucial for the overall validation of such techniques in our field and how we move forward to balance the biomolecular demand while maintaining collection integrity for the future.

Decision letter (RSOS-192034.R0)

05-Feb-2020

Dear Dr Evin

On behalf of the Editors, I am pleased to inform you that your Manuscript RSOS-192034 entitled "Building three-dimensional models before destructive sampling of bioarchaeological remains: a comment to Pálsdóttir et al (2019)" has been accepted for publication in Royal Society Open Science subject to minor revision in accordance with the referee suggestions. Please find the referees' comments at the end of this email.

The reviewers and handling editors have recommended publication, but also suggest some minor revisions to your manuscript. Therefore, I invite you to respond to the comments and revise your manuscript.

- Ethics statement

- Data accessibility

<http://datadryad.org/submit?journalID=RSOS&manu=RSOS-192034>

- Competing interests

- Authors' contributions

All submissions, other than those with a single author, must include an Authors' Contributions section which individually lists the specific contribution of each author. The list of Authors should meet all of the following criteria; 1) substantial contributions to conception and design, or

acquisition of data, or analysis and interpretation of data; 2) drafting the article or revising it critically for important intellectual content; and 3) final approval of the version to be published.

- Acknowledgements

- Funding statement

Because the schedule for publication is very tight, it is a condition of publication that you submit the revised version of your manuscript before 14-Feb-2020. Please note that the revision deadline will expire at 00.00am on this date. If you do not think you will be able to meet this date please let me know immediately.

- 1) A text file of the manuscript (tex, txt, rtf, docx or doc), references, tables (including captions) and figure captions. Do not upload a PDF as your "Main Document";
- 2) A separate electronic file of each figure (EPS or print-quality PDF preferred (either format should be produced directly from original creation package), or original software format);

- 3) Included a 100 word media summary of your paper when requested at submission. Please ensure you have entered correct contact details (email, institution and telephone) in your user account;
- 4) Included the raw data to support the claims made in your paper. You can either include your data as electronic supplementary material or upload to a repository and include the relevant doi within your manuscript. Make sure it is clear in your data accessibility statement how the data can be accessed;
- 5) All supplementary materials accompanying an accepted article will be treated as in their final form. Note that the Royal Society will neither edit nor typeset supplementary material and it will be hosted as provided. Please ensure that the supplementary material includes the paper details where possible (authors, article title, journal name).

If your manuscript is newly submitted and subsequently accepted for publication, you will be asked to pay the article processing charge, unless you request a waiver and this is approved by Royal Society Publishing. You can find out more about the charges at <https://royalsocietypublishing.org/rsos/charges>. Should you have any queries, please contact openscience@royalsociety.org.

on behalf of Professor Matthew Collins (Associate Editor) and Jon Blundy (Subject Editor)
openscience@royalsociety.org

Associate Editor Comments to Author (Professor Matthew Collins):
Dear Allowen

I am all three referees are in agreement that your comment should be published, and I am recommending this course of action to the editor, although he might ponder on the exact nature of your 'response'.

Some minor editing comments from me
Missing c in bioarchaeological "submitted to the same variety of..."
"100 euros per hour (less than 50 euros in our case) which translate" add an s translates

“ fetal ageing, sexing and dietary isotope studies” add space isotope studies
 missing from Acknowledgement Acknowledgement

Referee 1

The paper “Building three-dimensional models before destructive sampling of bioarchaeological remains: a comment to Pálsdóttir et al (2019)” provides clear and valuable information about the costs and time needed to make 3D models of bioarchaeological remains before destructive sampling and will be a useful resource for scientists in the field. The paper should be accepted with minor revisions.

Page 1

Line 27: The paper “Not a limitless resource: ethics and guidelines for destructive sampling of archaeofaunal remains” addresses more than just the archaeozoological community but also museums and other institutions that curate archaeofaunal collections as well as the disciplines doing destructive sampling such as the ancient DNA community. You might want to change the first line of the paper to reflect that.

Page 2

Line 3-4: The text is a bit unclear, I suggest changing it to something like this:

In terms of time, both approaches require working on a specimen for ~15 minutes, not including time to set up equipment (plus computer time for photogrammetry) but those methods only capture the external, visible, part of the object.

Line 18-19: The text is a bit unclear, I suggest changing it to something like this:

Depending on the resolution and the size of the specimens, this cost may change but would always be a fraction of the cost per sample for e.g. ancient DNA studies.

Line 22-24: The text is a bit unclear, I suggest changing it to something like this:

A recent paper demonstrates a correlation between X-ray dose-levels higher than 2000 Gy and decreasing aDNA quantities but no effect was detected for doses below 200 Gy [4].

Line 24: You talk about Gy and Gray, I am not familiar with this measurement unit, is it the same? If so the same spelling should be used if not the difference needs to be explained.

Line 33: I suggest changing this to

Petrous bones are now the first choice for many ancient DNA analyses because its density means it more often contains well preserved DNA [2,3], however because of the small size of the petrous bone, it is frequently completely destroyed during destructive analyses [1].

Referee 2

This is an unusual response paper, in that it's not reacting against anything in the original article, but rather expanding usefully on a point that Palsdottir et al make in passing - namely the potential of 3D scanning of zooarchaeological bones prior to destructive analysis. As such, it doesn't represent an essential correction or counterpoint, but is nonetheless a highly valuable contribution that is best read alongside the original paper. Hence I would recommend publication.

Referee 3

This is a very straight-forward commentary on the use of digital imaging for research and conservation of natural history collections.

There is no main question to report on with statistical results, but rather a reiteration of the value that imaging techniques have for the integrity of a collection that has requests for destructive sampling. The examples of future research are appropriate for conveying the crucial necessity of modelling/ scanning before destructive sampling.

The authors comment on aDNA degradation via X-ray which is important, though there is no mention of using a copper filter which can help prevent degradation. There are other issues to consider such as scan resolutions and quality of CT (for example to study trabecular structure in bone), this poses the question about how many scans should be allowed, one or multiple, depending on the research question or on the collection preservation?

As noted in the text, 3D models/scans are much easier to access, however, data storage of scans/models is not so simple or affordable to sort out, especially if you have a large collection to work with. I'm actually curious to know the thoughts of the authors on this - though I realise this may be beyond the scope of their comment.

Above all, the conclusion offers an important message about what data has been lost by not imaging remains before destruction and this has impact that I believe is worth putting in print.

Reviewer comments to Author:

Reviewer: 1

Comments to the Author(s)

Paper review:

The paper "Building three-dimensional models before destructive sampling of bioarchaeological remains: a comment to Pálsdóttir et al (2019)" provides clear and valuable information about the costs and time needed to make 3D models of bioarchaeological remains before destructive sampling and will be a useful resource for scientists in the field. The paper should be accepted with minor revisions.

Page 1

Line 27: The paper "Not a limitless resource: ethics and guidelines for destructive sampling of archaeofaunal remains" addresses more than just the archaeozoological community but also museums and other institutions that curate archaeofaunal collections as well as the disciplines doing destructive sampling such as the ancient DNA community. You might want to change the first line of the paper to reflect that.

Page 2

Line 3-4: The text is a bit unclear, I suggest changing it to something like this:

In terms of time, both approaches require working on a specimen for ~15 minutes, not including time to set up equipment (plus computer time for photogrammetry) but those methods only capture the external, visible, part of the object.

Line 18-19: The text is a bit unclear, I suggest changing it to something like this:

Depending on the resolution and the size of the specimens, this cost may change but would always be a fraction of the cost per sample for e.g. ancient DNA studies.

Line 22-24: The text is a bit unclear, I suggest changing it to something like this:

A recent paper demonstrates a correlation between X-ray dose-levels higher than 2000 Gy and decreasing aDNA quantities but no effect was detected for doses below 200 Gy [4].

Line 24: You talk about Gy and Gray, I am not familiar with this measurement unit, is it the same? If so the same spelling should be used if not the difference needs to be explained.

Line 33: I suggest changing this to

Petrous bones are now the first choice for many ancient DNA analyses because its density means it more often contains well preserved DNA [2,3], however because of the small size of the petrous bone, it is frequently completely destroyed during destructive analyses [1].

Reviewer: 2

Comments to the Author(s)

This is an unusual response paper, in that it's not reacting against anything in the original article, but rather expanding usefully on a point that Palsdottir et al make in passing - namely the potential of 3D scanning of zooarchaeological bones prior to destructive analysis. As such, it doesn't represent an essential correction or counterpoint, but is nonetheless a highly valuable contribution that is best read alongside the original paper.

I have only two suggestions for changes, both of which are minor:

1. Talking about costs is useful, if likely to become out-of-date fast. However, it seems a little odd to switch between \$ and €. I would recommend standardising currencies to one or the other.

2. In the conclusion you refer to destructive sample requirements becoming smaller and smaller until eventually they are not destructive at all. While it's theoretically possible that new technologies might permit entirely non destructive analyses in future, this would surely require lateral developments over and above the expected reduction in sample size requirements. I'm not sure such advances can be considered 'likely'. Hence I'd suggest replacing 'completely' with 'virtually'.

(nb. some extant "non-destructive" methods, e.g. the ambic soaking protocol for ZooMS, are in reality destructive, despite not having an immediately visible impact on sample morphology. Future 'non-destructive' methods should likewise be treated with scepticism until their long-term curatorial impacts are assessed)

Reviewer: 3

Comments to the Author(s)

Thank you for a straight-forward assessment of 3D modelling and scanning techniques that can be used for zooarchaeological remains. The emphasis on loss of data from remains not imaged prior to destruction is crucial for the overall validation of such techniques in our field and how we move forward to balance the biomolecular demand while maintaining collection integrity for the future.

Author's Response to Decision Letter for (RSOS-192034.R0)

See Appendix A.

Decision letter (RSOS-192034.R1)

24-Feb-2020

Dear Dr Evin,

It is a pleasure to accept your manuscript entitled "Building three-dimensional models before destructive sampling of bioarchaeological remains: a comment to Pálsdóttir et al (2019)" in its current form for publication in Royal Society Open Science. The comments of the reviewer(s) who reviewed your manuscript are included at the foot of this letter.

Please ensure that you send to the editorial office an editable version of your accepted manuscript, and individual files for each figure and table included in your manuscript. You can

send these in a zip folder if more convenient. Failure to provide these files may delay the processing of your proof. You may disregard this request if you have already provided these files to the editorial office.

on behalf of Professor Matthew Collins (Associate Editor) and Jon Blundy (Subject Editor)
openscience@royalsociety.org

Appendix A

Dear editor,

We have made the revisions suggested by the referees and reviewers.
We have uploaded two versions of the manuscript, one with all changes highlighted in yellow and a 'clean' version incorporating the changes.

Answered to reviewers and editors are listed below :

-Missing c in bioarchaeological "submitted to the same variety of..."

Done

-"100 euros per hour (less than 50 euros in our case) which translate" add an s translates

Done

-" fetal ageing, sexing and dietary isotope studies" add space isotope studies

Done

-c missing from Acknowledgement Acknowledgement

Done

Page 1

Line 27: The paper "Not a limitless resource: ethics and guidelines for destructive sampling of archaeofaunal remains" addresses more than just the archaeozoological community but also museums and other institutions that curate archaeofaunal collections as well as the disciplines doing destructive sampling such as the ancient DNA community. You might want to change the first line of the paper to reflect that.

The text has been changed accordingly

Pálsdóttir et al. [1] have recently encouraged bioarchaeologists, zooarchaeologists, museums and other institutions that curate archaeofaunal collections to recognise the importance of achieving reasoned destructive sampling strategies when performing analyses requesting destruction of bioarchaeological remains. Though the results obtained from approaches such as C14 dating, ancient DNA, palaeoproteomics, collagen fingerprinting or isotope analyses produce major advances in our understanding of past fauna and their relationships with humans, such approaches cannot (yet) be performed without the destruction of at least a small portion of bone or tooth.

Page 2

Line 3-4: The text is a bit unclear, I suggest changing it to something like this:

In terms of time, both approaches require working on a specimen for ~15 minutes, not including time to set up equipment (plus computer time for photogrammetry) but those methods only capture the external, visible, part of the object.

The text has been modified exactly as requested

Line 18-19: The text is a bit unclear, I suggest changing it to something like this:

Depending on the resolution and the size of the specimens, this cost may change but would always be a fraction of the cost per sample for e.g. ancient DNA studies.

The text has been modified exactly as requested

Line 22-24: The text is a bit unclear, I suggest changing it to something like this:

A recent paper demonstrates a correlation between X-ray dose-levels higher than 2000 Gy and decreasing aDNA quantities but no effect was detected for doses below 200 Gy [4].

The text has been modified accordingly with a slight modification :

A recent paper demonstrates a correlation between accumulating X-ray dose higher than 2000 Gray and decreasing aDNA quantities, but no effect was detected for doses below 200 Gray

Line 24: You talk about Gy and Gray, I am not familiar with this measurement unit, is it the same? If so the same spelling should be used if not the difference needs to be explained.
Agree, we have changed the text to Gray (Gy is the symbol).

Line 33: I suggest changing this to
Petrous bones are now the first choice for many ancient DNA analyses because its density means it more often contains well preserved DNA [2,3], however because of the small size of the petrous bone, it is frequently completely destroyed during destructive analyses [1].
The text has been modified and now appears as: Petrous bones are now the first choice for many ancient DNA analyses because the density of this bone means it reliably contains well preserved DNA.

Referee 2

This is an unusual response paper, in that it's not reacting against anything in the original article, but rather expanding usefully on a point that Palsdottir et al make in passing - namely the potential of 3D scanning of zooarchaeological bones prior to destructive analysis. As such, it doesn't represent an essential correction or counterpoint, but is nonetheless a highly valuable contribution that is best read alongside the original paper. Hence I would recommend publication.

We agree that is not a response but a comment (as mentioned in the title of the manuscript). We believe that supporting the content of an article can be just as important and useful as contradicting it.

Referee 3

The authors comment on aDNA degradation via X-ray which is important, though there is no mention of using a copper filter which can help prevent degradation. There are other issues to consider such as scan resolutions and quality of CT (for example to study trabecular structure in bone), this poses the question about how many scans should be allowed, one or multiple, depending on the research question or on the collection preservation?

As noted in the text, 3D models/scans are much easier to access, however, data storage of scans/models is not so simple or affordable to sort out, especially if you have a large collection to work with. I'm actually curious to know the thoughts of the authors on this - though I realise this may be beyond the scope of their comment.

We modified a paragraph to answer this important comment.

Because X-ray dose is cumulative caution should be made if multiple scans have to be performed on the same specimen (e.g. for multiple studies) and a possible safe recommendation would be to avoid scanning a given archaeological sample more than 5 times and to record the number of experiment conducted as well as the setting parameters used to perform the scan. Storing data has a cost (50 Gb of raw and derived data per hour in our case) which may be a limiting constraint if a large collection of 3D models has to be maintained.

Reviewer 2

I have only two suggestions for changes, both of which are minor:

1. Talking about costs is useful, if likely to become out-of-date fast. However, it seems a little odd to switch between \$ and €. I would recommend standardising currencies to one or the other.

Done

2. In the conclusion you refer to destructive sample requirements becoming smaller and smaller until eventually they are not destructive at all. While it's theoretically possible that new technologies might permit entirely non destructive analyses in future, this would surely require lateral developments over and above the expected reduction in sample size requirements. I'm not sure such advances can be considered 'likely'. Hence I'd suggest replacing 'completely' with 'virtually'.

Done

We also included some small changes to improve the clarity of the text. They are also highlighted in yellow.

Thanks a lot
Allowen Evin